# Death Notification in Italian Critical Care Unites and Emergency Services. A Qualitative Study with Physicians, Nurses and Relatives

**DOI:** 10.3390/ijerph182413338

**Published:** 2021-12-18

**Authors:** Ines Testoni, Erika Iacona, Lorenza Palazzo, Beatrice Barzizza, Beatrice Baldrati, Davide Mazzon, Paolo Navalesi, Giovanni Mistraletti, Diego De Leo

**Affiliations:** 1Department of Philosophy, Sociology, Pedagogy and Applied Psychology (FISPPA), University of Padova, 35122 Padova, Italy; erika.iacona@unipd.it (E.I.); lorenza.palazzo@unipd.it (L.P.); beatrice.barzizza@gmail.com (B.B.); beatrice.baldrati@tiscali.it (B.B.); davide.mazzon@unipd.it (D.M.); 2Emili Sagol Creative Arts Therapies Research Center, University of Haifa, Haifa 3498838, Israel; 3Anesthesia and Intensive Care Unit, University Hospital of Padova, 35128 Padova, Italy; paolo.navalesi@unipd.it; 4Department of Surgery and Medicine, University of Padua, 35128 Padova, Italy; 5Department of Medical-Surgical Pathophysiology and Transplantation, University of Milano, 20142 Milano, Italy; giovanni.mistraletti@unimi.it; 6Australian Institute for Suicide Research and Prevention, Griffith University, Brisbane 4122, Australia; d.deleo@griffith.edu.au

**Keywords:** COVID-19, death notification, critical care units, qualitative research, death education

## Abstract

This qualitative study was conducted in critical care units and emergency services and was aimed at considering the death notification (DN) phenomenology among physicians (notifiers), patient relatives (receivers) and those who work between them (nurses). Through the qualitative method, a systemic perspective was adopted to recognise three different categories of representation: 23 clinicians, 13 nurses and 11 family members of COVID-19 victims were interviewed, totalling 47 people from all over Italy (25 females, mean age: 46,36; SD: 10,26). With respect to notifiers, the following themes emerged: the changes in the relational dimension, protective factors and difficulties related to DN. With respect to receivers, the hospital was perceived as a prison, bereavement between DN, lost rituals and continuing bonds. Among nurses, changes in the relational dimension, protective factors and the impact of the death. Some common issues between physicians and nurses were relational difficulties in managing distancing and empathy and the support of relatives and colleagues. The perspective of receivers showed suffering related to loss and health care professionals’ inefficacy in communication. Specifically, everyone considered DNs mismanaged because of the COVID-19 emergency. Some considerations inherent in death education for DN management among health professionals were presented.

## 1. Introduction

In the area of breaking bad news, DN is the delivery of the news inherent in the death of a relative or friend to a designated individual (receiver). The notifier is the person who delivers the death notice and can be medical personnel or law enforcement. DN should be completed in a private space in a work office or at a hospital, and DN by telephone should be avoided because it results in impersonal communication [1,2].

DN is highly stressful for both the notifier and receiver. On one hand, it has the power to influence and contribute to the way receivers and their families cope with the grieving process [3]. A study carried out on 54 family members of 48 patients who died in intensive care or emergency departments showed that the factors most related to the outcome of bereavement were the attitude of the notifier, the transparency of the message, the confidentiality of the information and the ability of the notifier to answer the questions of the bereaved [4]. Studies have highlighted that empathic ways of supporting notification of the patient’s death are fundamental to the grief response in families and their satisfaction with medical staff. On the contrary, a cold and detached type of communication may increase distress and anxiety, emotions forever linked to the memory of a moment of rupture in one’s narrative history [5,6,7]. Deaths occurring in critical care units (CCUs) and emergency services (ESs) often risk being poorly communicated and, therefore, would be expected to increase the risk of a complicated grief process for receivers and their families [8]. On the other hand, DNs emotionally drain notifiers. Iserson’s study found that 70% of emergency physicians find informing survivors personally difficult [9]. The literature shows that physicians suffer from high stress responses when breaking bad news and an increase in heart rate when they have to deliver DNs when compared with a control group delivering good news [10]. Despite all this making this population more susceptible to dissatisfaction with job performance and duties, unfortunately, physicians do not receive specific training for improving their DN communication skills, and this deficiency facilitates burnout [8]. Death education is provided for help in this task because DN should make the notifiers sufficiently resilient and confident in their communication skills and make the survivors fully aware of the circumstances surrounding the death [5]. In the area of breaking bad news, there are different models and guidelines: for example, SPIKES [11], ABCDE [7], BEREAKS [6] and GRIEV_ING [12]. The common approach of these models is to build a positive relationship with the receiver before communicating bad news. However, despite the importance of this issue, the literature confirms that there are insufficient death education curricula in health profession degree courses [13,14]. During the first phases of the COVID-19 pandemic, DN in CCUs and ESs was particularly difficult because of the exorbitant number of sudden and traumatic deaths. To humanise the moment of separation, some hospitals have made a call or video call before the death of the patient [15]. During the COVID-19 pandemic, the Italian Society of Anaesthesia and Intensive Care (SIAARTI) association together with the Italian Society of Critical Care Nurses (ANIARTI), the Italian Society of Palliative Care (SICP) and the Italian Society of Emergency Medicine (SIMEU) associations, which include physicians, nurses working in intensive care, emergency and palliative care units, collaborated to produce a document: “CommuniCoViD: How to communicate with families living in complete isolation” [16]. The document presented instructions for communicating with family members in isolation and for communication via computer or telephone. The document recommended adopting a truthful approach, using unambiguous words appropriate to the receivers’ capacity for understanding, and an empathic space for emotional reactions. Although studies highlight the general need for the amelioration of breaking bad news, there are still no specific studies considering the importance of DN during the COVID-19 pandemic and how notifiers and receivers have managed this significant passage of the caring process.

## 2. Qualitative Research

### 2.1. Aims

The research aimed to give space to the narratives, experiences and accounts of those who had to give the DN (physicians), those who had to receive it (relatives of the deceased) and those who supported this interaction (nurses). With respect to the notifiers, we wanted to investigate how the experience of DN was lived, how the relationship with the receivers changed with respect to the previous modalities and what new strategies were implemented. With respect to the receivers, the study aimed to highlight their perceptions of DN and subsequent emotions. With respect to the nurses, we wanted to understand to what extent they were involved in the communication dynamics, what role they played and what effect their involvement had in their profession and private life.

### 2.2. Methods

This study adopted a qualitative research design [17], and snowball sampling was employed to involve participants. This non-randomised method is widely used in qualitative research, especially when members of a particular population are difficult to reach [18]. A presentation meeting was organised that involved supervisors of intensive care units in hospitals located in the same cities from which the authors came to ask managers to spread the communication to operating unit colleagues from different Italian regions who respected the study criteria. Speaking about receivers, all started from researchers’ acquaintances and social communications (Facebook), asking anyone to spread the word to other people who lived their same experience. The participants who took part in the research were all interested in participating. Participants came from all over Italy; in particular, 40 were from the North, 6 from the Central and 1 from the South. A total of 47 people took part (25 females and 22 males, mean age = 46.36; SD = 10.26), including 23 clinicians: 22 anaesthetists and resuscitators and one cardiologist (9 females and 14 males, mean age = 46.87; SD = 7.53); 13 nurses working in intensive care units (6 females and 7 males, mean age = 44.08; SD = 9.24); 11 family members of COVID-19 victims (10 females and 1 male, mean age = 48; SD = 15.7). Participant recruitment ceased when themes presented by participants became repetitive and the data achieved theoretical saturation. Two specially trained psychological interviewers conducted the interviews, and two experienced psychologists and one professor (all co-authors of this article) supervised the interview process. The semi-structured interviews were inspired by interpretative phenomenological analysis (IPA) [19,20] and realised through the internet, recorded and transcribed verbatim. Each interview lasted about 60 min. The researchers’ primary concern was to collect data from participants on the themes under investigation while considering the different perspectives of participants. The analysis aimed to recognise the main themes, as is characteristic of thematic analysis (TA) [21]. Similarly, as in other studies that integrate IPA and TA [22,23,24,25], the texts underwent an analysis to identify similarities and specificities across all the narratives [26]. The analysis was supported by Atlas.ti software [27].

The study respected the American Psychological Association’s Ethical Principles and Code of Conduct and was approved by the Ethics Committee of the University of Padua (n. 8DD829A1F8F83852FEDB64AAE38A4F79). Participants were informed about the study’s aims and procedures, and they were assured that participation was voluntary and that their responses would remain anonymous. To protect the participants’ identities, the names used in this text are pseudonyms.

## 3. Results Inherent to Physicians

This section may be divided into subheadings. It should provide a concise and precise description of the experimental results and their interpretation, as well as the experimental conclusions that can be drawn.

### 3.1. First Area of Thematic Prevalence: Changes in the Relational Dimension

This first area is divided into two sub-themes: relationships with patients and relationships with family members.

#### 3.1.1. Relationships with Patients

The closure of intensive care units to the relatives’ visits was perceived as a terrible limitation, as Gaia narrated: “We can know the patients through their relatives, and their narrations that describe us their story, transforming a passive ill subject into a person. This was absolutely impossible during Covid and our patients were totally depersonalised”. The use of sanitary facilities forced people to keep their distance, as Holly stated: “It became a relationship filtered by masks, by layers of dressing, it was more difficult to manage the emotional and motivational part of the patient”. Participants perceived to have been stigmatised as “death bearers”, as Ilaria said: “The first wave we felt excluded: you were in the Covid ward so you were not supposed to go near anyone, you couldn’t breathe. It is understandable, of course, but it is not that we are not human”. Finally, the reduction of the distance could be perceived as problematic, as Emanuele stated: “I felt a greater involvement at an emotional level towards patients and relatives, because I identified myself with them. I wasn’t able to balance distance and closeness”.

#### 3.1.2. Relationships with Family Members

The use of internet devices partially reduced the distance between relatives and patients. As Olga said, “Digital technology helped us in reducing the relational distance. The ability to make video calls was a relief because it gave both the patient and the family member direct contact again”. However, distancing critically worsened the relationships with relatives, as Carlo said: “Before the Covid, it was already difficult to make the family understand the health situation of their relatives, despite they could see them. During the pandemic it was absolutely impossible”. Further difficulties were related to the language and culture barriers with immigrated people, as Emanuele said: “As many simple words as we could use in some cases it was evident that they were not effective. Where before gestures, tone, expressions could help, on the phone it was almost impossible”. Finally, the daily communications with relatives were perceived as exhausting, as Andrea affirmed: “There were at least 30 relatives to notify every day. Taking three hours a day to communicate with them was really challenging”.

### 3.2. Second Area of Thematic Prevalence: Difficulties Related to Death Notification

Social distancing creates many difficulties, especially in the DN process. Ilaria said, “I was always used to the fact that the patients died surrounded by their relatives and there was always the last goodbye. During the pandemic, I have had patients who have left home saying, ‘I’m going to the hospital for a moment’ and never said goodbye. How to inform relatives that they died?”. Marco also affirmed: “Usually the relatives are conscious of the situation when they see a loved one entering the intensive care unit, because they can see that their beloved is to dying... If they cannot see the situation, it’s hard to help them. When we said that we were going to stop the treatments, the most typical reaction was ‘keep at it a bit longer’, so it is very difficult to make the notification”. The sudden death of patients made it impossible to prepare family members in advance, as Holly described: “It was very difficult when the death came suddenly. One minute you’re saying goodbye to the family and they’re stable and suddenly they’re worse within a short space of time. In those cases, we had a great sense of surprise on the other side and often relatives became inquisitive”. Paolo also said, “I couldn’t cope with this chain of deaths any longer; having to communicate deaths is very tiring and emotionally demanding”. All participants underlined the necessity of specific training in communication, like Lorenzo: “I would like to have a training also made with a practical part, with simulations. I also believe that it is necessary to create culture, to sensitise doctors and nurses”. Carlo also said, “There is an absolute and vital need to spread what we already know about how to communicate the death of people; this is fundamental. They are not innate skills; they are learned and perfected through study, experience and in-depth study”.

### 3.3. Third Area of Thematic Prevalence: Distress and Protective Factors

The third area is composed of two sub-themes: causes of distress and protective factors.

#### 3.3.1. Causes of Distress

Informational and logistical disorders created massive distress as Natalie said: “I think a lot of the stress was initially caused by having confused information, for example not knowing how to protect yourself”. Olga further underlined, “We didn’t know if the devices and procedures we were using were correct, if the treatments were effective or harmful, so it was a moment of genetical uncertainty”. The excessive workload was described by all participants. Francesco, for example, said: “We were absolutely improvised, in operating blocks with insufficient and exhausted staff. We were never off work and came home exhausted. We slept very little and then we went back to the hospital again, another 12 or 14 h of exhausting work”. The extremely high number of deaths was psychologically unbearable, as Francesco said: “It was a deep suffering. There was pain upon pain; you didn’t have time to get rid of one when another one came and then another one. What could you say when you had two, three deaths a day all the time?”. Among the causes of distress were the moral decisions (moral distress) related to the management of limited resources, as Ilaria said: “In the first wave, if we had 50 patients and 20 places, we would ask ourselves ‘Who am I to decide who lives and who dies?’”.

#### 3.3.2. Protective Factors

The most important protective factor was the support of loved ones, as Marco said: “Having a wife who understood, with whom I could talk and vent was vital. Even with my eldest son, we spoke on the phone a lot. They have been a great resource.” Cohesion among colleagues was equally significant, as Olga affirmed: “What helped us was realising that behind the suit, behind the masks we were not alone, but the colleague working next to me was experiencing the same emotions as me and so we were able to give strength to each other”. Spirituality was a personally useful resource, as Gaia described: “Spirituality gave me great comfort, in the most difficult moments I felt that I would be able to overcome the situation that until the night before seemed unresolvable”.

## 4. Results concerning the Nurses

The thematic areas which emerged from the nurses’ narrations were changes in the relational dimension, the impact of the death and protective factors.

### 4.1. First Area of Thematic Prevalence: Changes in the Relational Dimension

#### 4.1.1. The Relationships with the Patients

The closure of intensive care units increases the difficulties in the nurse/patient relationship as well, as Rachele said: “Because family members also help us to know the patients, so we know better who they are, what they prefer, what they like to do, their beliefs. So, we were missing all this”. The use of contagious protection devices also hindered contact with patients, as Jane affirmed: “These devices reduce sensitivity. With the double gloves, even when you touch the patients, you don’t touch them as if you were touching them with a thread of glove that is like your hand. So felt impotent because I wasn’t being able to have a good care relationship”. Finally, the fear of contagion enlarged the distance, as Stefano reported: “If the person you assist scares you, it’s the old story of the AIDS patient in the 90s, isn’t it? If you are afraid of the person you are caring for, you cannot treat him well, because you keep him at a distance and suppress the relationship”.

#### 4.1.2. The Use of Videocalling

Some nurses described the benefits of technology as Kevin: “So knowing what’s going on, the emotion, the expression is important, and that’s what you can do with technology, with a simple video call”. Rachele, however, highlighted the difficulties of this instrument: “I was hindered at the beginning by the technology, because I wasn’t able to use it”. Tommaso underlined the difficulties related to cultural differences in managing video calling: “Cultural differences created some difficulties in communication with video calling. Italians are much more aseptic in communication or in the manifestation of their emotions, while other cultures manifest emotions in a more visible way. All this created some difficulties in the maintenance of a proper distance”.

### 4.2. Second Area of Thematic Prevalence: The Impact of Death

#### 4.2.1. The Drama of High Death in Loneliness

Social distancing created considerable difficulties in dealing with death. As Walter reported, “To die like this, alone, as it happened during the Covid period to thousands of people, unfortunately, in my opinion is a dramatic thing”. Quentin also says that “this situation went on throughout the whole pandemic period, they didn’t have family around, they had no religious assistance, they were dying and they were completely alone”. Dying alone and the high number of victims had a high emotional impact on the participants.

#### 4.2.2. Death notification

Nurses did not often participate in DN, but when they did, the impact was high. Participation in DN produced acute stress. Video calls were the only way to say goodbye to the patient. Tommaso recounted that: “They were certainly powerful experiences, especially if you think of the person on the other side who may not have seen their loved one for a month and then sees them again at the end of life. So, it was certainly powerful. I still maintain that it was a good thing and that if the relatives felt like it, it was really the only way to make them say goodbye”.

### 4.3. Third Area of Thematic Prevalence: Protective Factors

The third area consists of two sub-themes: relational resources and post-traumatic growth.

#### 4.3.1. Relational Resources

The protective factors were those described by physicians. Jane said, “What helped me was that my relatives never hindered, they never saw me as a danger in the house. They’ve never said to me ‘we’re afraid you’re going to take us in”. Furthermore, cohesion with colleagues helped them, as Yuri affirmed: “Certain colleagues were definitely morally supportive, often even physically supportive and we definitely helped each other”.

#### 4.3.2. Post-Traumatic Growth

Despite all difficulties, the emergence in some cases improved the personal growth that Ursula described as follows: “I also brought out a lot of myself that I didn’t think I could bring out, or that I had this strength inside me, and in fact I rediscovered myself as a person”. Quentin also recognises that he has learned a lot from this experience: “It has certainly toughened us up a lot and also tightened the ranks, those who have come out of this experience like us, have come out of it much stronger from a professional point of view”.

## 5. Results Concerning the Receivers

As for the group of victims’ relatives, the themes that emerged were the following: The hospital being perceived as a prison, bereavement between DN, lost rituals and continuing bonds.

### 5.1. First Area of Thematic Prevalence: The Hospital Perceived as a Prison

This first area was divided into two sub-themes: the effect of social distancing and the reclusion of ill relatives and relationships with health professionals.

#### 5.1.1. The Effects of Social Distancing and the Reclusion of the Ill Relative

Not being able to stay close to the patient caused deep anguish, as Maria reported: “I suffered a lot receiving the death notification because my mum had to die with her hands clasped in mine. Or I would have wanted to tell her, ‘Mum, we will see each other again one day. I will choose you from a thousand mums. You will always be my mum. You were precious to me’. When I heard that she had died, I began to suffer for not having told her this”. Imagining the sick relatives alone in the hospital without being able to visit them as if they were in prison created a lot of anxiety and guilt. Cristina said, “What I don’t accept is that I feel I haven’t done enough, or that I haven’t helped him in this battle he had to face alone, not even having an ally”. Luca said, “It’s already terrible to lose a loved one even when you can accompany them. When you cannot be near someone you love who is suffering and you can only communicate with a tablet, it is terrible. I wouldn’t wish this on anyone.”

#### 5.1.2. Relationships with Health Professionals

Some receivers described a supportive and helpful attitude, as Lorena said: “When I was contacted by the first doctor, I felt very welcome, he explained things to me and also told me that their activity acted as a psychological support for the family members. Then things changed and two or three days before my father’s death, I went to look for that doctor who had called me the first time”. Indeed, many difficulties mined the relationships with healthcare staff. The first one was the staff turnover: “The doctors changed all the time and each doctor gave me a different interpretation, some saw improvements and others told me there was nothing to do” (Angela). “There was always a different nurse speaking. We didn’t have a contact person, and we received different information. That’s why my brother and I decided to go to the hospital to ask what was happening to our father” (Marta). Some participants reported feeling that they were treated in a cold and unempathetic manner, as Irene stated: “A doctor brutally told my mother ‘it’s obvious that he’s not going to make it’. My mother was very upset. With the doctors there was never any emotional communication related to the fact that there would be a detachment from this person”. Then, the care given to their relatives was perceived as inadequate and dehumanising.

### 5.2. Second Area of Thematic Prevalence: Bereavement between Death Notification, the Lost Ritual and Continuing Bonds

The second area of thematic prevalence is divided into two sub-themes: the experience of DN and the bereavement.

#### 5.2.1. Death Notification

Receivers perceived a lack of clear information about the circumstances of the death, as was the case for Maria: “After the death notification, the director of the department, instead of rolling up his sleeves, closed the offices, he was nowhere to be found. He abandoned everyone!” Participants, like Lorena, often described DN as cold and distancing: “He was cold, detached. Certainly, he was reporting a death, so he was probably in discomfort, I don’t doubt it, but cold and distant”. Anna reported a similar experience: “A doctor told me: ‘Ah madam, I was just about to call you, to tell you that your father is dead’. In a state of disbelief, I asked: ‘And now?’ Answer: ‘And now, wait for us to call you because we’re going to do a swab to see if your father is positive or not, then we’ll tell you”. Generally, reactions were shock and derealisation, as is the case with Luciana: “I couldn’t keep this inside me, I didn’t shed a single tear. In my opinion, those were all lies. I didn’t believe what he had told me”. From the perception of being abandoned, the desire for justice by taking legal action was derived, as narrated by Cristina: “Although I want him in jail because he did not have the capacity to manage such a vulnerable area. However, I preferred to attack the administration. I want to see which protocol was followed, because I want to see it. There are too many inconsistencies! They were wrong from the start!”.

#### 5.2.2. Bereavement

Bereavement was perceived as particularly difficult because of derealisation: “I still cannot understand the fact that my mother is dead because when you can’t see it, it’s hard to realise. It’s a strange feeling, I don’t realise it’s happened” (Marta). Some narrations showed a deep sense of guilt: “I had a not wonderful relationship with my father. I wanted to tell him ‘I love you’ before he died and I couldn’t, so now I feel guilty” (Cristina). “I feel that I gave Mum the impression of having abandoned her, despite it wasn’t possible for me to be near her” (Maria). Generalised anxiety and cognitive deficits were reported by several participants, such as Luciana: “I had problems with my head, with my memory I never remembered anything. At that moment I don’t know what happened, but I was always crying, at home I used to put the music on and then take it off, I forgot things”.

#### 5.2.3. Lost Ritual and Continuing Bonds

A further difficulty was caused by the impossibility of celebrating the funeral, as in Flavia case: “Since there was no funeral, it’s as if something was missing, as if the circle hadn’t been closed. I’m not a very religious person, but I missed it so much”. The treatment of the corpse according to the hygienic norms shocked relatives, as Lorena explained: “We made the tragic discovery of what the protocols are for the Covid dead. They didn’t have the possibility to be dressed in a funeral before they had to be enclosed in a rubbish bag. They were sealed in two bags”. The reshaping of the usual funeral rituals helped to manage the grief, as Anna described: “I felt a sense of relief when a ceremony was performed. The formula is not that of a funeral, but for us, it was vital. A ritual had somehow been done, and helped to manage grief”. The continuing bonds seemed to support participants, as Luciana narrated: “I talk to him every day, in the morning and in the evening when I go to bed, I greet him and say ‘sleep well’. I feel better when I talk to him and also when I talk about him because, in this way, I know that he is alive”. However, some experiences of the deceased intrusion into the participants’ present were described, as narrated by Cristina: “I try to remove the idea that he is not there. I don’t dream about him, but I see him everywhere. I see him outside my shop, I see him in other people’s faces, I see him everywhere”, and by Maria: “I still dream of my mother every night, every night. Maybe because she is still part of me, I can’t let her go, I don’t know. On the one hand I’m happy because I feel her so close to me, but on the other hand I think I’m hurting myself by forcing her to stay with me, I don’t know”.

Finally, participants also acknowledged that they felt a benefit from the interview, as Stefano reported: “I wanted to thank you and thank you for this kind of journey, because I believe that it is necessary that, among the various things we will understand in the next 20 years regarding this Covid theme, we also try to understand ourselves a little”.

## 6. Discussion

The study considered the context in which the DN happened, seeing the relationships between notifiers, receivers and those who work between them: the nurses. As already considered by the literature [28], with respect to the relationships of physicians and nurses with patients, all the issues were pivoted on pandemic treatment modifications that were dramatic and undermined the previous consolidated relational schemes. In particular, before the pandemic, CCUs wards were used to working with “open doors” to welcome family members at any time of the day [29]. This innovation, also widely adopted in Italy, has ensured a significant improvement in care relationships [30,31,32]. This innovation runs in parallel with the previous formation in the person-centred approach that characterises the most recent evolution of CCU intervention [33,34]. Having to prevent the physical presence of relatives had a significant emotional impact on both physicians and nurses, leading them to feel like executioners [35,36]. The closure of the wards and the absence of relatives contributed to the level of depersonalisation, which was aggravated by the use of sanitary facilities that forced people to keep their distance. Despite these conditions, communication mediated by phone and computer were perceived as solutions to the communication problem; these devices imposed a huge transformation in relational dynamics, causing many misunderstandings and did not help in the construction of the therapeutic alliance. As observed in the literature [15], despite it being extremely hard for them, nurses tried to make the video calls when they recognised the last moments of the patient’s life and felt it was their duty to grant a last goodbye, but all these experiences deeply hurt them. Difficulties were related to the language, the impossibility of explaining everything, lack of intimacy, lack of cultural mediators, and scarce ability in the use of the devices.

With respect to the DN, when it was impossible to contact relatives for the last goodbye by tablet, further complications were accounted for by physicians, mainly related to the ability to communicate empathically. Our study agreed with the literature that the impossibility of permitting relatives to see their loved ones was an obstacle to preparing them for the death of the patient [37]. Furthermore, it was almost impossible to achieve good compliance with stopping treatment because of worries about becoming aware of the inevitability of death. Furthermore, communication by internet or telephone did not remain as private as in normal conditions [38]. Some nurses who participated in the DN experienced a negative emotional impact. Then, DN resulted in a very stressful and painful event. Physicians underscored the necessity of receiving adequate communication training to better manage these situations. All this was made more difficult by the contextual situation, which was perceived as disorganised and unpredictable, and by the excessive workload that was aggravated by daily communications with families. A high number of deaths, a sense of impotence and frustration, traumatic experiences, anxiety, fear and fatigue constantly undermined the mood of the doctors who had to give the DN. Physicians were asked to make life-and-death decisions and suspend or select treatments because of limited resources, and then they had to give the DN after the effect of these intense experiences of moral injury. Both physicians and nurses experienced the pandemic as a severely stressful and terrible event for which they were unprepared [39,40]. Only solidarity among colleagues and the support of a loved one could help physicians overcome this dramatic phase. For some of them, spirituality and religiosity are further protective factors, as already underlined by the literature [28]. However, almost all nurses had a desire to move away from the traumatic experience with the hope of not losing the lessons learned. Several stressed the fact that they had changed for the better professionally and personally and that they had discovered unexpected parts of themselves and humanity that they did not want to lose. These aspects suggest post-traumatic growth, which consists of the experience of positive change following an adverse experience [41,42].

With respect to the receivers, the context of the DN was extremely difficult to manage. Only a few participants described a supportive and helpful relationship with health professionals who had been able to help them maintain positive contact with their loved ones. The hospital was generally perceived as a prison where their beloved one was secluded. The social distancing caused intense anguish and a sense of guilt due to the fear of having abandoned the deceased during the final moments of their existence. All this impacted the effect of the DN and the subsequent experiences of grieving without counting on social support. Given this background, the relationships with healthcare professionals resulted in difficulties and misunderstandings, and the care offered to their beloved was perceived as inadequate and dehumanising, which developed concern regarding the dying moments. The lack of clear information about the circumstances of the death and the practices carried out by the health professionals during treatment created a sense of bewilderment and anger. Cold and unempathetic manners paralleled with experiences of mismanagement and lack of clarity regarding the circumstances of death caused few receivers to take legal action. The emotions ranged from anger to anxiety and a sense of guilt for having been abandoned or for having abandoned their loved one, frequently highlighting the presence of unresolved suspicions that mismanagement could have caused the death. Some factors had a detrimental effect on communication: the staff turnover prevented the construction of an ongoing relationship and thus the possibility of establishing an empathic dialogue and a therapeutic alliance sufficient to accept the DN. Uncertainty about the development of the illness and expectations of death were constantly perceived as latent content that should be notified at any moment, but when the bad news arrived, it was shocking or seemed unrealistic. An intense concern about the treatment of the corpse according to the hygienic norms of the burial protocols was intertwined with the sensation that everything did not happen. All of this caused difficulty in working on loss because experiences were often shrouded in a sense of unreality, as already described by the literature [43]. The impossibility of seeing the dead body and having contact with it intensified this sensation. The common areas that unify the narrations of all three groups are as follows. First, the open hospital is confirmed as an excellent strategy to positively improve doctor/patient/family relationships. Before the pandemic, the relative’s inclusion permitting 24-h visits, participation in mortuary toilets, pets’ visits, and family lounges significantly enhanced the quality of patients’ stay in hospital and the relational dimension between health care professionals, patients and families [44]. The COVID-19 crisis stopped all this and added further barriers: the fear of the virus, equipment-related constraints, exacerbated communication difficulties and an extreme lack of medical time [44]. The second common aspect is inherent to the confirmation that communication by phone or computer is not adequate for DNs. Unfortunately, even in cases of a full-blown emergency, such as a pandemic, receivers harbour resentment and anger at the way this vital information is given, despite being aware of the inevitable difficulties that the COVID-19 challenge imposes. Although in the first phase of the COVID-19 outbreak the literature positively emphasised the use of the telephone and computer as a means of ensuring proximity between patients and their families [45,46,47,48], communication between healthcare professionals and receivers has been ineffective in terms of reassurance and understanding. Unfortunately, there are no studies that checked the follow-up of this kind of communication, which, in our study, results in not-so-positive outcomes.

## 7. Conclusions

DN was a highly negative experience for all participants in this research because of the context in which it occurred and the exceptional way in which it was handled. Many factors play a role in this consequence. Among them, we can list the inexperience in the use of the phone and computer medium, the stress caused by the urgency, the lack of adequate models to manage the breaking bad news and the DN in critical situations. Both physicians and nurses underlined the importance of improving their communication skills, but despite the importance of this issue, as confirmed by the literature [49], no studies have considered how to manage this lack of competencies.

Finally, the interviewees not only welcomed participation in the project, but many wanted to say a few words about the interview, highlighting the importance of being able to share their experiences so that they could be reworked and re-signified, despite the pain resulting from the memory. Studies in the literature confirm that raising awareness of the upsetting experience to construct a meaning that can be integrated with the personal biography allows the creation of a meaningful narrative rather than a traumatic one and favours readjustment to normal life [50,51].

### Limits and Future Studies

The study did not analyse the issue of perceived stigmatisation. Future research should better consider how health professionals manage this perception and its origin. Other studies should better develop a follow-up on the effects of computer- and telephone-mediated communication in dealing with breaking bad news and DN. This study would allow for the development of an appropriate and useful communication model in emergencies similar to the COVID-19 pandemic. Two further significant aspects we found and did not sufficiently consider so they require future study: professionals had the perception to be stigmatised as “death bearers”, while the absence of relatives and other figures on the ward has led professionals to take global responsibility for the needs of patients without being able to delegate anything. This reduced the necessary distance with patients and relatives, producing too-close involvement and further distress.

## Data Availability

The datasets used during the current study are available from the corresponding author on reasonable request.

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
