# Peer review of "Death Notification in Italian Critical Care Unites and Emergency Services. A Qualitative Study with Physicians, Nurses and Relatives"

_ijerph, 2021, doi:10.3390/ijerph182413338_

Round 1

Reviewer 1 Report

The article is well organized and the methodology adopted is appropriate since it is a qualitative study. However, one notices the total lack of analysis that goes beyond the transcription of a few responses deemed particularly significant. Giorgio or Natalie's statements are probably congruent with the expectations of the researchers, but how many of the interviewees expressed similar content? The research is purely descriptive and lacks any starting hypotheses whose proof is derived from the data collected. The contribution is not particularly relevant to the increase of scientific knowledge about the notification of death: it ranks more among the pilot studies than among the qualitative research in the strict sense.

Author Response

We thank you sincerely for the revision of our article “Death Notification in Italian Critical Care Unites and Emer-gency Services. A qualitative study with physicians, nurses and relatives”. 

Reviewer 1 – Comments/Requests: The article is well organized and the methodology adopted is appropriate since it is a qualitative study. However, one notices the total lack of analysis that goes beyond the transcription of a few responses deemed particularly significant. Giorgio or Natalie's statements are probably congruent with the expectations of the researchers, but how many of the interviewees expressed similar content? The research is purely descriptive and lacks any starting hypotheses whose proof is derived from the data collected. The contribution is not particularly relevant to the increase of scientific knowledge about the notification of death: it ranks more among the pilot studies than among the qualitative research in the strict sense.

Authors’ response: 

The present study aims at investigating how death notification was managed during Covid-19 emergency and how healthcare professionals managed this crucial step in the care process during such a complex time nobody was ready to face.

The starting points were not expectations or hypothesis to confirm, since it is not a step established in the qualitative method; we wanted to give a platform to the storytelling of the ones who had to give death notifications and the ones who received those news, in order to learn more about their experiences, how the communication process changed during the pandemic, which strategies were used and - starting from what was observed and the critics arisen - develop a communication model that could be useful and suitable for emergencies similar to the Covid-19 pandemic.

Reviewer 2 Report

This paper presents an interesting qualitative study about experience with death notifications among Italian physicians, nurses and family members of COVID-19 patients. They are important since Italy was one of the country that was hit hardest in COVID-19 pandemics.  It needs however, a small clarification in methodology section. The uathros state that participants came from all over Italy  and that They used snow balling sampling methods. Maybe a bot more description regarding sample could improve the methodology section. How were the participants approached  and did all participants immediately accepted to participate in the interviews. From which Italian regions did the participants come.

Author Response

We thank you sincerely for the revision of our article “Death Notification in Italian Critical Care Unites and Emer-gency Services. A qualitative study with physicians, nurses and relatives”. 

Reviewer 2 – Comments/Requests: This paper presents an interesting qualitative study about experience with death notifications among Italian physicians, nurses and family members of COVID-19 patients. They are important since Italy was one of the country that was hit hardest in COVID-19 pandemics.  It needs however, a small clarification in methodology section. The authors state that participants came from all over Italy  and that They used snow balling sampling methods. Maybe a bot more description regarding sample could improve the methodology section. How were the participants approached and did all participants immediately accepted to participate in the interviews. From which Italian regions did the participants come.

Authors’ response: We have included the required information in the methods section (lines 110-118).
